# The Impact of Molecular and Genetic Analysis on the Treatment of Patients with Atypical Meningiomas

**DOI:** 10.3390/diagnostics14161782

**Published:** 2024-08-15

**Authors:** Janez Ravnik, Hojka Rowbottom

**Affiliations:** Department of Neurosurgery, University Medical Centre Maribor, 2000 Maribor, Slovenia; hojka.rowbottom@ukc-mb.si

**Keywords:** atypical meningioma, genetics, adjuvant radiotherapy, recurrence, treatment plan

## Abstract

Meningiomas represent approximately 40% of all primary tumors of the central nervous system (CNS) and, based on the latest World Health Organization (WHO) guidelines, are classified into three grades and fifteen subtypes. The optimal treatment comprises gross total tumor resection. The WHO grade and the extent of tumor resection assessed by the Simpson grading system are the most important predictors of recurrence. Atypical meningiomas, a grade 2 meningioma, which represent almost a fifth of all meningiomas, have a recurrence rate of around 50%. Currently, different histopathologic, cytogenetic, and molecular genetic alterations have been associated with different meningioma phenotypes; however, the data are insufficient to enable the development of specific treatment plans. The optimal treatment, in terms of adjuvant radiotherapy and postoperative systemic therapy in atypical meningiomas, remains controversial, with inconclusive evidence in the literature and existing studies. We review the recent literature to identify studies investigating relevant atypical meningioma biomarkers and their clinical application and effects on treatment options.

## 1. Introduction 

Meningiomas account for around 40% of all primary tumors of the central nervous system (CNS) in adults, with 4.2 cases per 100,000 surgically relevant cases per year [1,2,3,4]. Based on their cytological features, meningiomas were thought to be derived from arachnoid cap cells; however, the exact origin remains unknown, with a possible origin being the arachnoid barrier cells, since they share the expression of prostaglandin D synthase with meningiomas [5,6,7]. The fifth edition of the World Health Organization (WHO) classification of CNS tumors classifies meningiomas into three grades and fifteen subtypes [8,9,10]. The extent of surgical resection, evaluated by Simpson grading of resection degree [11], and the CNS WHO grade are the most important predictors of the recurrence risk of meningiomas, with approximately 80% being CNS WHO grade 1, which have a favorable prognosis [12]. The advances in molecular profiling have emphasized the role of genetic and epigenetic changes in characterizing meningiomas and their aggressiveness and recurrence risk [13]. Currently, there are no clinically validated liquid biopsy prognostic or diagnostic biomarkers specific for meningiomas, which would enable a non-invasive diagnosis of meningiomas and their subtypes [14].

In CNS WHO grade 2 meningiomas, the recurrence rate is approximately 50%; thus, the European Association of Neuro-oncology (EANO) guidelines recommend radiotherapy or observation of patients with CNS WHO grade 2 that have undergone gross total tumor resection [15,16,17,18]. The 5-year progression-free survival (PFS) and overall survival rates for atypical meningiomas are 48–68% and 78–91%, respectively [19,20]. 

CNS WHO grade 2 meningiomas represent approximately 18.3% of all meningiomas [1]. According to the latest WHO classification of CNS tumors from 2021, grade 2 meningiomas can be further divided into three subtypes, chordoid, clear cell, and atypical, with the latter being the most frequent [10]. Although the female sex is recognized as a risk factor for meningiomas, higher-grade meningiomas are more common among men [1,21,22]. Patients with grade 2 meningiomas are often younger than those with grade 1 [21]. The biological characteristics of atypical meningiomas are between benign and anaplastic meningiomas [23]. 

Atypical meningiomas are diagnosed based on the following criteria: a mitotic index ranging between 4 and 19 mitoses per 10 fields of 0.16 mm^2^ and/or brain invasion and/or at least three minor criteria, which are spontaneous necrosis, pattern-less architecture (sheeting), small cells with a high nuclear/cytoplasmic ratio, macronucleoli, and hypercellularity [8,10]. The diagnostic criteria for atypical meningiomas are presented the Table 1. Atypical meningiomas have high heterogeneity and invasiveness [23]. Magnetic resonance imaging (MRI) characteristics of atypical meningiomas are an uneven signal, irregular borders with no clear boundary with the surrounding brain tissue, large areas of oedema, a generally large neoplasm volume, or a significant increase in size over a short time [24,25]. For grade 3 meningiomas, which account for 1 to 5% of all meningiomas and are more prevalent among men, the diagnostic criteria comprise a mitotic index of 20 or more mitoses per 10 fields of 0.16 mm^2^, frank anaplasia (sarcoma, carcinoma, or melanoma-like appearance), or papillary/rhabdoid histology. Histologic subtypes of grade 3 meningiomas are anaplastic, papillary, and rhabdoid meningiomas. TERT promoter mutations or homozygous deletion of cyclin-dependent kinase inhibitors A and B (CDKN2A/B) are molecular features of grade 3 meningiomas [10,26]. 

In cases of atypical meningiomas, the tumor has to be removed as completely as possible, including the dura and the base of the tumor as well as the invaded skull [27]. Grade 2 meningiomas are most often found within the convexity, whereas those in the skull base, however rare, are often smaller, less aggressive and, in the majority of cases, located in the sphenoid locations [28,29,30]. Gross total resection is linked with a longer overall survival, and postoperative radiotherapy is offered to patients with subtotal resection of atypical meningiomas [20,31]. The role of adjuvant radiotherapy in atypical meningiomas after gross total resection (GTR) remains controversial, with some studies demonstrating durable local control without adjuvant radiotherapy, while others advocate radiotherapy in cases of atypical meningiomas, even with GTR [32,33,34]. 

## 2. Histopathological Features 

Brain invasion, characterized by the infiltration of tumor cells into the brain parenchyma without intervening leptomeninges, is, according to the fifth WHO classification of CNS tumors, enough to classify a meningioma as atypical [10]; however, studies have shown that atypical meningiomas diagnosed on the sole presence of brain invasion and lacking other criteria have a recurrence rate similar to that of CNS WHO grade 1 meningiomas [35,36,37]. In cases of tumor reappearance, other features associated with a higher risk of recurrence, such as a high Ki-67 labeling index, incomplete surgical removal, or occurrence in the context of neurofibromatosis type 2 (NF2), were present, suggesting that the relapse was independent of brain invasion [36]. Brain invasion in otherwise benign meningiomas is associated with a lower probability of recurrence compared to the high mitotic activity present in an atypical meningioma [8]. A meta-analysis of 25 studies including 3560 atypical meningiomas found no difference in the progression risk between tumors with a mitotic index of four or more mitoses per ten high-power fields (HPFs) or below that cut-off [38]. In comparison, a study with 200 atypical meningiomas discovered that a cut-off of six mitoses per ten HPFs was connected to recurrent cases [15]. Other studies have suggested that eight mitoses per ten HPFs are associated with progression in cases of atypical meningiomas [8,39]. Some propose the Ki-67 index as an alternative to the mitotic index to evaluate the proliferation of meningiomas [8]. In a cohort of 99 atypical meningiomas, a Ki-67 index above 7.5% was recognized as a significant predictor of shorter relapse-free survival (RFS) [38]. As the Ki-67 index can be affected by inter-observer variability, no universally accepted threshold has been established to distinguish between high and low proliferation to recognize atypical meningiomas with a greater risk of recurrence [8]. A high Ki-67 index is associated with increased neoplasm malignancy and an average index of 3.8% is mostly found in benign meningiomas and 7.2% in atypical meningiomas; thus, high Ki-67 expression is linked with relapse after surgery (Table 2) [40]. 

Atypical meningiomas, diagnosed based on minor atypical criteria, have a significantly lower risk of recurrence [15,41]. In cases of atypical meningiomas with major and minor criteria, in some studies, sheeting, macronucleoli, and spontaneous necrosis were recognized as indicators of recurrence [15,42], whereas an analysis of 262 atypical meningiomas and a meta-analysis of 25 studies found that spontaneous necrosis was the only significant prognostic factor among minor criteria for progression [38]. 

Immunocytochemistry staining enables the distinction of atypical meningiomas from gliomas, neurogenic tumors, mesenchymal tumors, and some metastatic tumors with epithelial membrane antigen (EMA), with vimentin being the preferred diagnostic marker for meningiomas [43,44]. Most meningiomas express EMA with a decreased expression index in high-risk meningiomas (WHO grade 2 and 3) and vimentin is also present in the majority of cases of meningiomas, especially in high-risk ones (grade 2 and 3) [23]. Somatostatin receptors (SSTRs), present in meningiomas, are involved in tumorigenesis, with SSTR2A being an independent prognostic factor regarding meningioma recurrence and with higher SSTR2A expression being linked to more aggressive meningiomas [45,46,47]. Furthermore, the expression of progesterone receptors in meningiomas is also linked to recurrence, with higher rates of progesterone receptors being associated with a better prognosis and a limited chance of recurrence or malignant transformation [48]. Additionally, high p53 expression and an elevated Ki-67 index are also features of more aggressive meningiomas [49]. 

## 3. Cytogenetic Features 

### 3.1. Copy Number Alterations

The progressive accumulation of chromosomal losses and gains leads to meningioma progression, and monosomy of chromosome 22q is believed to be an early event in the pathogenesis of meningiomas in both NF2-mutated and non-NF2-mutated meningiomas [50,51,52,53,54,55]. 22q loss is present in many patients with an established NF2 mutation, and the somatic mutation is present in approximately 47% of sporadic meningiomas [56]. In WHO grade 2 meningiomas, additional chromosomal copy number aberrations (CNAs), such as a loss of 1p, 14q, 18q, 10q, and 6q and a gain of 20q, 12q, 15q, 1q, 9q, and 17q, are present (Table 2) [50,57]. A loss of chromosomal locations 1p and 14q has been linked to high-grade meningiomas, and 1p loss is recognized as the second most common chromosomal event after 22q loss and is associated with higher rates of tumor recurrence and progression [58,59,60]. 

In grade 1 meningiomas, which later progressed, cytogenetic abnormalities associated with grades 2 and 3 were present, and thus, CNAs could be used for the identification of low-grade meningiomas with a higher risk of recurrence and progression [59,61]. A study where the whole-genome CNAs were analyzed in patients with atypical meningiomas discovered that 3.5 CNAs were linked to recurrence [62]. Specific chromosomal CNAs in atypical meningiomas could be used as prognostic markers of progression and recurrence [8]. 10q loss, found using next-generation sequencing (NGS), was recognized as an indicator of a more aggressive meningioma [63]. Additionally, 1p loss was the most significant predictor of RFS [64]. A greater number of chromosomal anomalies are associated with a higher-grade meningioma [65]. Changes in genes HIST1H1C and CTGF, located on chromosome 6, are linked with meningioma recurrence and a loss of heterozygosity at certain sites of chromosome 10, and can be indicators of a worse prognosis in cases of meningiomas [66]. 

### 3.2. Molecular Models for Risk Assessment 

Based on this knowledge, a molecular model for risk assessment was proposed [8]. Meningiomas, bare psammomatous, angiomatous, secretory, and clear cell, should be assessed for 1p deletion, and cases lacking this CNA are to be considered low risk of progression; however, when 1p deletion is present, analysis for 6q and 14q losses has to be conducted, and with two losses, a meningioma is considered high risk [64]. This risk stratification model was successful in categorizing the recurrence risk in cases of atypical meningiomas [67]. In cases of atypical meningiomas, an association was discovered between 1p and 10q loss and shorter RFS, with 7p and 18 losses also being recognized as negative prognostic factors [68]. It has therefore been suggested that cases of atypical meningiomas without 1p deletion can be managed conservatively [64,68]. A heterozygous deletion of cyclin-dependent kinase inhibitors A and B (*CDKN2A/B*) was also associated with a shorter time of meningioma recurrence, and it importantly correlates with shorter RFS in surgically resected atypical meningiomas [41,69]. 

### 3.3. Cytogenetic Features in Post-Radiation Meningiomas 

Meningiomas represent the most common neoplasm type to appear after radiotherapy, with a tendency to be more aggressive than their sporadic counterparts, and a loss of 1p was found to play an important role in the development of radiation-induced meningiomas, followed by changes in chromosomal locations 9p, 19q, and 22q [70,71,72,73]. Ionizing radiation is the sole environmental risk factor linked with the development of meningiomas and there seems to be a genetic susceptibility to radiation-induced meningiomas [74]. 

## 4. Molecular Genetics and Classification of Meningiomas 

The latest CNS WHO classification has not changed the criteria for diagnosing atypical meningiomas; however, for the first time, specific genetic alterations have been included in the grading, such as TERT promoter (*p*TERT) mutations and/or *CDKN2A/B* homozygous deletions (HDs), which are associated with meningioma recurrence and progression [12,69,75,76,77,78,79]. A meta-analysis of 677 patients with WHO grade 2 meningiomas found *p*TERT mutations in 7.9% and another analysis of 63 atypical meningiomas found *p*TERT mutations in only 1.6% [68]. So far, *p*TERT mutations have often been present in secondary atypical meningiomas, suggesting that primary and secondary atypical meningiomas have different molecular pathways [80]. Similarly, a study of 183 atypical meningiomas identified *CDKN2A/B* HD in a mere 4% of WHO grade 2 meningiomas [76]. 

Mutations in the genes AKT1, SMARCB1, and SMO (Table 2) are rare in cases of atypical meningiomas, and mutations in the genes KLF4 and TRAF7 are associated with an indolent clinical behavior, thus making these meningiomas have a low risk of progression [54,68,80,81,82]. SMO, a G-coupled receptor, is important in the Hedgehog signaling pathway, which is crucial in promoting angiogenesis and tumor progression, and is linked with larger meningiomas and a higher recurrence rate than that in AKT1-mutated meningiomas [83,84,85]. Mutations in PIK3CA (Table 2), which encodes for a catalytic subunit of phosphatidylinositol 3-kinase (PI3K), can be found in cases of atypical meningiomas and often co-occur with TRAF7 mutations, especially in cases of skull base meningiomas [86,87]. Changes in SUFU also lead to the dysregulation of the Hedgehog signaling pathway, which is important for meningioma growth and development, and they often co-occur with PTEN and ARID1A mutations (Table 2) [52,88]. 

### 4.1. The Role of NF2 Gene Mutations in Meningiomas 

NF2 gene mutation is associated with the early development of meningiomas [89,90,91] and is recognized as the most common gene mutation in cases of atypical meningiomas (Table 2), present in up to 75% of tumors, making a meningioma 3.78 times more likely to be atypical than without an NF2 gene alteration [68,80,92]. NF2 loss-of-function mutation is caused by a double-hit mechanism in meningiomas, either by a germline mutation and a second hit with a somatic mutation in syndromic cases or with a somatic single-nucleotide variation, or by an insertion/deletion mutation and an overlapping chromosome 22 deletion, which is commonly seen in sporadic cases [93]. Neurofibromatosis type 2, now termed NF2-related schwannomatosis, caused by a mutation in the NF2 gene, results in the loss of function of Merlin, a tumor suppressor protein, which leads to the overexpression of YAP1 (yes-associated protein 1) [94,95,96]. The transcriptional co-activator YAP1 has an important role in tissue development and homeostasis, and the activation of YAP1 is linked with the loss of function of a tumor suppressor NF2/Merlin, which enables tumor growth, invasion, and resistance to apoptosis [97,98,99,100,101,102,103]. YAP1 fusion meningiomas resemble low-grade NF2 mutant meningiomas; based on their upregulated genes and their downregulated genes, they resemble high-grade NF2 mutant meningiomas [94,100]. Hyperactive YAP1, a consequence of the YAP1 fusion protein binding to the TEA domain, leads to tumor cell proliferation and invasion, playing an oncogenic role in meningioma tumorigenesis [98,104,105]. 

### 4.2. Epigenetic Modifications

The presence of immunohistochemical loss of histone H3 trimethylation at lysine 27 (H3K27me3), a DNA epigenetic modification, could be used as an indicator of an increased risk of tumor recurrence and negative prognosis, and it is associated with grade 2 meningiomas with shorter disease-free survival (Table 2) [8,106,107]. Additionally, cases of atypical meningiomas after stereotactic surgery can imply that these tumors are more resilient to radiation therapy [108]. In a recent study of paired samples of primary meningiomas and relapsed cases, H3K27me3 loss was present in 35% of relapses and 25% of cases following radiotherapy, thus linking the epigenetic modification to adjuvant treatment and tumor reoccurrence [108,109]. The role of H3K27me3 immunohistochemical loss needs to be further investigated, and in some studies, the results of staining are ambiguous and difficult to interpret in a clinical setting [109]. The Table 2 lists the most important factors for the recurrence of atypical meningiomas.

Patel et al. identified three classes of meningiomas (type A, B, and C), based on bulk RNA sequencing and whole-exome sequencing, with type C having the highest proliferative index, the shortest RFS, and an increased expression of FOXM1, leading to a loss of the repressive DREAM complex [81]. Vasudevan et al. also divided meningiomas into two types, with increased expression of FOXM1 being an indicator of an aggressive meningioma, and in primary atypical meningiomas, FOXM1 was often upregulated [80,110]. 

### 4.3. Genomic Analysis-Based Meningioma Division 

Aggressive meningiomas can be divided into three groups based on genomic analysis of their NF2 status: NF2 mutant, NF2 agnostic, and NF2 wild-type. NF2 mutant are mostly associated with men and mutations in *CDKN2A/B*, whereas NF2 agnostic are linked with TERT and TP53 mutations [52]. 

Genome-wide DNA methylation profiling on 497 meningiomas performed by Sahm et al. further divided meningiomas into six methylation patterns for the prediction of progression and recurrence [111]. MC ben-1 to 3 had a benign course, MC int-A and B had an intermediate behavior, and MC mal was associated with an aggressive course. Atypical meningiomas were generally classified as MC int-A and MC int-B; however, some were also classified as MC ben-1 (96,97). Nassiri et al. used copy number variation analysis, somatic point mutations, methylation profiles, and messenger RNA abundance of 121 meningiomas, creating four molecular subtypes, immunogenic, benign NF2 wild-type, hypermetabolic, and proliferative, allowing them to better predict the outcomes compared to the 2016 WHO classification of CNS tumors [112]. According to the methylation analysis conducted by Olar et al., two groups of meningiomas were distinguished with different prognoses; a higher proportion of copy number aberrations was present in the unfavorable prognosis group, including losses of 1p and 14q [113]. 

### 4.4. Integrated Molecular–Morphological Grading System of Meningiomas 

Combining the DNA methylation patterns, chromosomal CNAs, and WHO grade, Maas et al. created an integrated molecular–morphological grading system of meningiomas, where each meningioma was scored based on their WHO grade (0–2 points), methylation class (0–4 points), and chromosomal loss of 1p, 14q, and 6q (0–3 points). Meningiomas scoring 0–2 points were classified as low risk, with 3–5 points as intermediate risk and those with 6–9 points as high risk [64]. By using the integrated molecular–morphological grading system, researchers did not need to evaluate pTERT and/or *CDKN2A/B* HD to estimate the recurrence risk [64,114]. 

Driver et al. also developed a grading system comprising mitotic counts, *CDKN2A/B* deletion, and chromosomal CNAs, where each CNA among 1p, 3p, 4p/q, 6p/q, 10p/q, 14q, 18p/q, and 19p/q deletions and *CDKN2A/B* deletions gained one point; one point was given for a mitotic index of 4 to 19 mitoses/1.6 mm^2^ and two points for a mitotic index of 20 or more mitoses/1.6 mm^2^. Meningiomas with 0–2 points were classified as grade 1, 2–3 points as grade 2, and 4 or more points as grade 3. By using the grading system, they were able to more accurately predict the risk of recurrence compared to the 2007 and 2016 WHO grading system [64,115]. 

Choudhury et al., in 2022, conducted DNA methylation profiling on 565 meningiomas, dividing them into three groups. The first one was merlin-intact meningiomas, characterized by a gain of function in chromosome 5, a loss of function in chromosome 6p, and intact NF2 expression, which had the best survival among the three groups. Second was an immune-enriched meningioma, characterized by a gain of function in 6p and a loss in 22q, with lymphocytes exhibiting exhaustion markers. The last group was hypermitotic meningiomas, with the worst overall survival, characterized by an upregulation of FOXM1 expression [116,117]. 

## 5. Management 

Initial treatment of atypical meningiomas comprises surgery with a maximal safe resection [118,119]. Simpson grades 1 to 3 are considered GTS and grades 4 to 5 a subtotal resection, with grade 0 representing complete tumor removal, with the removal at 2 to 3 cm from the tumor insertion site with good results [120,121]. The surgical approach has to be wide enough to allow for meningioma exposure with its dural attachments whilst enabling the surgeon to visualize the surrounding structures to disrupt the blood flow and minimize brain retraction and manipulation [122]. Small and asymptomatic meningiomas, believed to be benign, can be monitored [123]. Around a fifth of meningiomas display aggressive tendencies, and radiation therapy can be utilized as well as repeated surgery [119]. Following gross total resection of an atypical meningioma, the 5-year recurrence rate is 29–58%, with the 5-year survival rate being 79–91% and the 10-year survival rate being 53% [124,125,126,127]. 

Surgical resection is linked with improved neurological function; however, GTR is not always attainable, depending on the meningioma’s location and the incorporation of nerves and blood vessels [43,128,129]. Radiation therapy is most often indicated as an adjuvant therapy when subtotal resection of a WHO grade 2 meningioma is performed, and it has to be personalized, based on the neoplasm’s size, proximity to critical structures, and prior radiation to the same site, with the goal being a reduction in tumor proliferation and control over its progress [16,130,131]. 

Radiation therapy can be delivered as conventional fractioned photon radiotherapy, stereotactic radiotherapy, stereotactic radiosurgery or fractional stereotactic radiation therapy, intensity-modulated photon radiation therapy, and particle therapy with photons or carbon ions [119,123,132]. External beam radiation therapy is recommended for grade 3 meningiomas, whereas small grade 2 meningiomas can be treated with stereotactic radiosurgery, especially when their diameter is less than 3 cm and they are more than 3 mm from radiosensitive tissues [125,133,134]. In cases of atypical meningiomas with residual tissue, stereotactic radiosurgery is generally used with doses ranging from 12 to 20 Gy [131,135,136]; however, a study conducted by Kano et al. discovered that patients who received less than 20 Gy had a PFS of 29% at 5 years, whereas those who received 20 Gy had a PFS of 63% [135]. In a phase II trial, intensity-modulated photon radiotherapy with doses ranging from 54 to 60 Gy in 30 fractions was administered to patients with incompletely resected atypical meningiomas, which demonstrated that the aforementioned treatment modality was safe and effective and required further research [137]. The administration of particle radiation therapy with either protons or carbon ions enables higher radiation doses, which could improve local tumor control; however, larger prospective studies are still required [138]. Figure 1 shows one example of a patient with an atypical meningioma treated surgically and with radiotherapy.

The role of radiation in the treatment of patients with atypical meningiomas remains controversial and is often based on physician preference since the literature on this topic is sparse, with the majority of conducted studies being small and retrospective with low power and often inconsistent results [126,139]. Arguments in favor of adjuvant radiation are the reduction in the recurrence risk, the increased time to recurrence, a smaller tumor burden in cases of recurrence, and an improvement in disease-specific survival [130,136,140]. On the other hand, the arguments against adjuvant radiotherapy are that it does not reduce the recurrence risk and exposes patients to unnecessary radiation with potential harm [131,141]. Some studies on the effects of adjuvant radiotherapy demonstrated lower recurrence rates, but the results were not statistically significant [126,140], whereas other studies showed no difference in recurrence rates to those of active monitoring [121,131,142]. Hasan et al. conducted a meta-analysis of 14 retrospective studies comparing patients with atypical meningiomas undergoing GTR alone with those that received GTR and adjuvant radiotherapy and discovered cases of recurrence that occurred 8 months later in the group treated with adjuvant radiotherapy [130]. Graffeo et al. also conducted a meta-analysis of seven studies including their data, comparing recurrence rates between patients with atypical meningiomas treated with GTR alone and those that after GTR received radiotherapy, and discovered a trend of lower 5-year recurrence rates and improved overall survival in patients in the radiation group; however, the differences were not statistically significant [126]. 

A phase II trial (RTOG 0539) was a prospective study comparing the outcomes of recurrence in atypical meningiomas after GTR with adjuvant radiotherapy using a dose of 54 Gy. In the study, the 3-year PFS was 93.8%, which was significantly higher than in historical controls, and the recurrence rate at 3 years was 4.1%, with a 96% overall survival rate [143]. Another prospective phase II trial (EORTC 22042–26042) compared the results after GTR of atypical meningiomas alone and those treated with radiotherapy using a high dose of 60 Gy, and after 3 years, the PFS was 90% and the overall survival rate was 96.4% [144]. 

The optimal radiation dose is illusive, with doses ranging from 50 to 60 Gy administered in 1.8–2.0 Gy fractions to the tumor bed and residual tumor tissue with a margin ranging from 0.5 to 2.0 cm [123,130]. Atypical meningiomas are mostly treated with 54 Gy; however, doses of up to 70 Gy have been used and several retrospective studies showed that higher doses may improve patients’ outcomes [145]. 

Systemic therapy has limited effects on meningiomas and should be considered when all surgical and radiotherapy options have been explored, since several studies conducted so far have shown that chemotherapeutic agents, namely temozolomide, doxorubicin, and vincristine, were not effective against meningiomas [123,146,147,148]. Hydroxyurea can be used in patients with atypical meningiomas when postoperative radiotherapy cannot be applied as there were no differences in the PFS period between hydroxyurea chemotherapy and radiotherapy after surgery [149]. It was discovered that hydroxyurea was linked with longer PFS of patients with atypical meningiomas after stereotactic radiotherapy than with only conservative treatment, and even following GTR, hydroxyurea led to stabilization or even shrinking of the atypical meningioma [149,150]. 

Interferon-alfa, an immunomodulating agent, can be administered in cases of recurrent meningiomas where repeat surgery is not an option, since several studies have demonstrated its effects of stabilization of tumor growth, with a slight improvement in PFS at 12 weeks, but with no effect on the overall survival [123,151]. 

Meningiomas often overexpress growth factor receptors, such as vascular endothelial growth factor (VEGF), platelet-derived growth factor (PDGF), and epidermal growth factor (EGF); therefore, several studies using monoclonal antibodies or small-molecule kinase inhibitors have been conducted with limited or no success [147,152,153,154]. Anti-VEGF molecules and mTOR inhibitors have shown potential in the treatment of high-grade and recurrent meningiomas where surgical and/or radiotherapy have been less effective [119]. VEGF plays an important role in the development of the peritumoral oedema in meningiomas, and some authors have found a connection between high levels of VEGF and recurrence [155], whereas others linked VEGF to histological grade [156]. However, some did not find a correlation between the two factors, and it currently stands that VEGF should not be used as a marker of histological grade [157,158,159]. Oedema, caused by VEGF, is related to increased morbidity, and thus, anti-VEGF therapy, such as bevacizumab, has been trialed; however, the results so far have been disappointing [152,154,160,161,162]. Sunitinib, a small-molecule kinase with different targets, has been tested on atypical meningiomas, and PFS after 6 months improved from 5–30% to 42% [152]. The mTOR protein complex, a part of the PI3K complex, is crucial for meningioma development, and the mTOR inhibitor everolimus has demonstrated an increase in PFS and a decrease in tumor growth rate [163,164]. Meningiomas often express progesterone receptors; however, high-grade meningiomas tend to express estrogen receptors, but currently, anti-estrogenic agents have not demonstrated a strong effect [119,165]. Aggressive meningiomas with high relapse rates exhibit overexpression of somatostatin receptors, but inhibitors have shown little effect so far [166]. In the future, microRNAs could also be included in meningioma treatment, since a high expression of miR-190a and a low expression of miR-29c-3p and miR-219–5p have been linked with higher recurrence rates, and miR-21 has been upregulated in cases of grade 2 and 3 meningiomas [167]. 

## 6. Clinical Application of Cytogenetic Features of Atypical Meningiomas

Deng et al. created recommendations for the molecular diagnosis and treatment of meningiomas with the help of the Group of Neuro-Oncology, the Society of Neurosurgery, the Chinese Medical Association, neuropathologists, and evidence-based experts, based on identifying and analyzing relevant studies already available [168]. Based on the literature, they recommended radiation therapy in cases of meningiomas with pTERT mutations, which are associated with radiation sensitivity in de novo high-grade meningiomas. In all other cytogenetic features, the role of radiation therapy was unknown. Additionally, the researchers recommend that the combination therapy of everolimus with bevacizumab or octreotide could be considered for PI3K-activated meningiomas, and that pharmacotherapy using bevacizumab targeting VEGF could only be considered if no further local treatment option exists. Sunitinib can be used in cases of 22q and 1p deletion [119,168]. The authors suggest that meningiomas with 22q deletion, 18q deletion, NF2 mutation, and MO or SUFU mutation should be followed annually for 5 years, cases with 1p deletion, 14q deletion, AKT1 or PIK3CA mutation, H3K27me3-negative meningiomas, and TIMP3 or TP73p methylation-positive meningiomas should be followed every 6 months for 5 years, meningiomas with pTERT mutations should be followed every 3 to 6 months indefinitely, and meningiomas with *CDKN2A/B* HDs should be followed every 6 months indefinitely [168].

## 7. Conclusions

Despite considerable progress in understanding molecular and genetic alterations in meningiomas, their clinical application in the treatment of atypical meningiomas is still low. The diagnostic criteria for those tumors are becoming more refined, but they have little impact on current clinical management strategies. Surgery remains the most efficient treatment option. There are currently no specific guidelines for when adjuvant radiotherapy should be introduced. Future prospective studies are expected to produce more relevant information about the clinical usefulness of molecular and genetic diagnostics in atypical meningiomas.

## Figures and Tables

**Figure 1 diagnostics-14-01782-f001:**
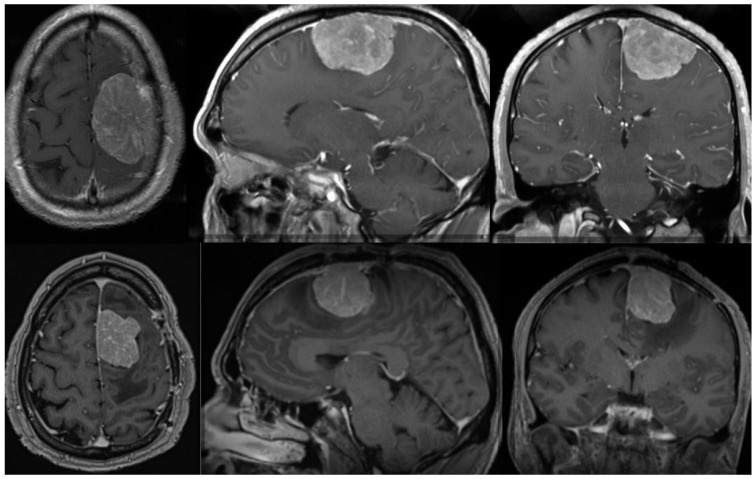
Preoperative (**upper row**) MR of the brain (axial, sagittal, and coronal view after contrast enhancement) showing a 51-year-old male with an atypical meningioma. The patient was operated on and the tumor was removed (Simpson grade 2 resection). The tumor exhibited brain invasiveness and 7 mitotic figures/10 high-power fields. The wall of the superior sagittal sinus, where the tumor was attached, was only coagulated and the sinus was preserved. The patient had no tumor recurrence 3 months after surgery and an 11 mm large recurrence was present after 9 months. (**Lower row**) The lower row shows the substantial recurrence of the tumor after 15 months. The patient was successfully reoperated on (Simpson grade 2 resection) and received adjuvant radiotherapy on the tumor bed. He had no obvious recurrence three months after the second surgery. The tumor was histologically identical after the second surgery. Genetic analysis was performed after both surgeries, revealing a mutation in the NF2 gene in both cases. No other chromosomal or important genetic abnormalities were detected.

**Table 1 diagnostics-14-01782-t001:** Diagnostic criteria for atypical (WHO grade 2) meningioma: fulfilling either 1 of 2 major criteria or 3 of 5 minor criteria.

**Major Criteria**
4–19 mitotic figures/10 high-power fields
Brain invasion
**Minor Criteria**
Increased cellularity
Small cells with high N/C ratio
Large and prominent nucleoli
Pattern-less or sheet-like growth (loss of lobular architecture)
Foci of spontaneous or geographic necrosis

**Table 2 diagnostics-14-01782-t002:** Factors for more likely recurrence of atypical meningiomas after surgery.

	Factor
Clinical	Incomplete removal (Simpson grade 2–4)
	Fast growth
Histopathological	High mitotic index/Ki-67
	Major diagnostic criteria
Chromosomal	Loss of 1p, 14q, 18q, 10q, 6q, 7p Gain of 20q, 12q, 15q, 1q, 9q, 17q
Genetic	NF2 mutation
	AKT1, SMARCB1, SMO, PIK3CA. SUFU, PTEN, ARID1A gene mutations
Epigenetic	H3K27me2 loss

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
