# Peer review of "The Impact of Molecular and Genetic Analysis on the Treatment of Patients with Atypical Meningiomas"

_diagnostics, 2024, doi:10.3390/diagnostics14161782_

Round 1

Reviewer 1 Report

Comments and Suggestions for Authors

The Authors address the clinically highly relevant topic of atypical meningioma diagnosis (histopathological and molecular) and treatment with emphasis on the impact of molecular features on prognosis and therapy. There has been several excellent reviews on genetics, and the current manuscript heavily relies on them. The added value is the neurosurgical/neurooncological viewpoint. 

Specific comment:

1) Discuss the difference between grade 2 and 3 meningiomas, both in terms of morphology and molecular features. T

2) Table 2: Include the relevant genetic findings of the 3 grading systems (discussed at 1.1).

3) Immunhistochemistry (lines 106-111): include and discuss SSTR2A and Progesteron receptor immunhistochemistry.

4) p53 immunhistochemistry (IHC) in combination with Ki67 and Progesteron receptor IHC has also predictive value regarding recurrence (although genetic methods have higher predictive value).

5) Line 43: chordoid (not choroid)

6) Line 27 and Line 53: Cite the WHO 5th Edition

Author Response

The Authors address the clinically highly relevant topic of atypical meningioma diagnosis (histopathological and molecular) and treatment with emphasis on the impact of molecular features on prognosis and therapy. There has been several excellent reviews on genetics, and the current manuscript heavily relies on them. The added value is the neurosurgical/neurooncological viewpoint.

Specific comment:

1) Discuss the difference between grade 2 and 3 meningiomas, both in terms of morphology and molecular features. T

Response: Thank you very much for your comment. We made corrections accordingly. We added morphological characteristics and molecular features of grade 3 meningiomas and their histologic subtypes to the penultimate paragraph of the Introduction, therefore, making a direct comparison with grade 2 meningiomas and their morphology and molecular features mentioned beforehand in the same paragraph.

2) Table 2: Include the relevant genetic findings of the 3 grading systems (discussed at 1.1).

Response: Thank you very much for your comment. We made corrections accordingly. We added additional chromosomal and genetic findings associated with atypical meningioma recurrence, which were already discussed in subheadings Cytogenetic features and Molecular genetics and classification of meningiomas in order to make the article more transparent for the reader as the text mentions numerous chromosomal, genetic, epigenetic and histopathological features linked with atypical meningiomas. Additionally, we linked the table content to the main text by using the table captions.

3) Immunhistochemistry (lines 106-111): include and discuss SSTR2A and Progesteron receptor immunhistochemistry.

Response: Thank you very much for your comment. We made corrections accordingly. At the end of the subheading Histopathological features, we discussed the role of SSTR2A and progesterone receptors on possible meningioma recurrence.

4) p53 immunhistochemistry (IHC) in combination with Ki67 and Progesteron receptor IHC has also predictive value regarding recurrence (although genetic methods have higher predictive value).

Response: Thank you for your comment. We made corrections accordingly and added the role of p53 and elevated the Ki-67 index in aggressive meningiomas.

5) Line 43: chordoid (not choroid)

Response: Thank you for your comment. Corrections were made accordingly.

6) Line 27 and Line 53: Cite the WHO 5th Edition

Response: Thank you for your comment. Corrections were made accordingly.

Reviewer 2 Report

Comments and Suggestions for Authors

The main question addressed by the review article «The impact of molecular and genetic analysis on the treatment of patients with atypical meningiomas» is current state of the relationship between the understanding of molecular and genetic changes in meningiomas and clinical use of existing knowledge. The authors discuss this problem in the context of atypical meningiomas.

The authors have described in detail the studies have shown the relationship between histopathological, cytogenetic, and molecular genetic changes and meningioma phenotypes, but conclude that these data are insufficient to develop optimal treatment for atypical meningiomas due to inconclusive data.

This is an interesting and important study that is relevant in this area. The paper is well written, the text clear and easy to read. The conclusions are consistent with the evidence; the references are appropriate.

This paper can be accepted for publication after minor revision.

1.         It would be nice to place links in table captions to the contents of the tables

2.         Is there data on research of noninvasive tumor markers of atypical meningiomas and their recurrences?

Author Response

The main question addressed by the review article «The impact of molecular and genetic analysis on the treatment of patients with atypical meningiomas» is current state of the relationship between the understanding of molecular and genetic changes in meningiomas and clinical use of existing knowledge. The authors discuss this problem in the context of atypical meningiomas.

The authors have described in detail the studies have shown the relationship between histopathological, cytogenetic, and molecular genetic changes and meningioma phenotypes, but conclude that these data are insufficient to develop optimal treatment for atypical meningiomas due to inconclusive data.

This is an interesting and important study that is relevant in this area. The paper is well written, the text clear and easy to read. The conclusions are consistent with the evidence; the references are appropriate.

This paper can be accepted for publication after minor revision.

  1. It would be nice to place links in table captions to the contents of the tables

Response: Thank you for your comment and suggestion. Corrections were made accordingly. We linked the table content to the main body of the article by using table captions.

  1. Is there data on research of non-invasive tumor markers of atypical meningiomas and their recurrences?

Response: upon specific search, we found a review article (Korte, B., Mathios, D. Innovation in Non-Invasive Diagnosis and Disease Monitoring for Meningiomas. Int. J. Mol. Sci. 2024, 25, 4195. https://doi.org/10.3390/ijms25084195) about the potential of a liquid biopsy and analysis of circulating tumor biomarkers (e.g., circulating tumor cells, circulating tumor DNA, microRNA, proteins, extracellular vesicles, epigenetic signatures) in bodily fluids, namely blood and cerebrospinal fluid, however, the authors state that currently there are no clinically validated liquid biopsy prognostic or diagnostic biomarkers specific for meningiomas. We included that data in the article at the end of the first paragraph of the Introduction.

Round 2

Reviewer 1 Report

Comments and Suggestions for Authors

The Authors have sufficiently answered my questions and addressed my queries. The manuscript has significantly improved at revision.